# Recent Development and Applications of Stretchable SERS Substrates

**DOI:** 10.3390/nano13222968

**Published:** 2023-11-17

**Authors:** Ran Peng, Tingting Zhang, Sheng Yan, Yongxin Song, Xinyu Liu, Junsheng Wang

**Affiliations:** 1College of Marine Engineering, Dalian Maritime University, Dalian 116026, China; ztt1999@dlmu.edu.cn (T.Z.); yongxin@dlmu.edu.cn (Y.S.); 2Institute for Advanced Study, Shenzhen University, Shenzhen 518060, China; shengyan@szu.edu.cn; 3Department of Mechanical and Industrial Engineering, University of Toronto, Toronto, ON M5S 3G8, Canada; 4Department of Information Science and Technology, Dalian Maritime University, Dalian 116026, China; 5Liaoning Key Laboratory of Marine Sensing and Intelligent Detection, Dalian Maritime University, Dalian 116026, China

**Keywords:** surface-enhanced Raman scattering (SERS), stretchable SERS substrate, flexible SERS substrate

## Abstract

Surface-enhanced Raman scattering (SERS) is a cutting-edge technique for highly sensitive analysis of chemicals and molecules. Traditional SERS-active nanostructures are constructed on rigid substrates where the nanogaps providing hot-spots of Raman signals are fixed, and sample loading is unsatisfactory due to the unconformable attachment of substrates on irregular sample surfaces. A flexible SERS substrate enables conformable sample loading and, thus, highly sensitive Raman detection but still with limited detection capabilities. Stretchable SERS substrates with flexible sample loading structures and controllable hot-spot size provide a new strategy for improving the sample loading efficiency and SERS detection sensitivity. This review summarizes and discusses recent development and applications of the newly conceptual stretchable SERS substrates. A roadmap of the development of SERS substrates is reviewed, and fabrication techniques of stretchable SERS substrates are summarized, followed by an exhibition of the applications of these stretchable SERS substrates. Finally, challenges and perspectives of the stretchable SERS substrates are presented. This review provides an overview of the development of SERS substrates and sheds light on the design, fabrication, and application of stretchable SERS systems.

## 1. Introduction

Surface-enhanced Raman scattering (SERS) was discovered in 1977 by Van Duyne [1] and Creighton [2] as an efficient, ultrasensitive, label-free analytical technique. Since its discovery, interest in and use of surface-enhanced Raman spectroscopy has grown exponentially, with attractive advantages in areas such as environmental analysis [3], food safety [4,5], and biomedical analysis [6,7]. However, the capability of this technology has not been fully developed due to many unexplored challenges such as the development of highly stable and sensitive SERS substrates [8,9,10,11,12]. The intensity of the Raman signal is affected by many factors, such as the size, shape, surface roughness of plasmonic nanostructures, etc., but the key is to locate the target molecules in the gap of the metal structures where the electromagnetic field is highly enhanced (i.e., the “hot spot”) [13]. The enhancement factor of SERS is proportional to the local electromagnetic field strength; thus, the gap distance of the metal nanoparticle determines the Raman hot-spot intensity, which in turn affects the performance of SERS devices [14].

Traditional SERS substrates are developed on rigid materials such as silicon and glass substrates with fixed hot-spot sizes. The sample loading efficiency for large-size targets into small hot-spots is low, which directly leads to unsatisfactory Raman detection sensitivity [15]. Flexible SERS substrates developed based on a flexible matrix, which enables conformable sampling on arbitrary surfaces, are suitable for in situ and on-site detection of targets [10,16,17]. The flexible sampling process can highly improve the detection feasibility, but still with limitations for detecting relatively large-size samples due to the limited hot-spot size [15]. Compared to traditional rigid and flexible SERS substrates, stretchable SERS substrates are prepared on stretchable supporting materials which can effectively improve the sample loading efficiency by increasing the distance between metal nanoparticles [18] and then enhance the Raman signal intensity by reducing the distance between metal nanoparticles [19]. The integration of stretchable materials with SERS substrates paves the way for the development of high-performance Raman sensors. However, research on the topic of stretchable SERS substrates is still limited.

This paper reviews the fabrication and applications of the emerging stretchable SERS substrates (see Figure 1). The working mechanism of SERS and the features of rigid, flexible, and stretchable SERS substrates are briefly overviewed. Stretchable SERS substrates prepared by in situ wet chemical synthesis, physical deposition, physical adsorption, embedding of plasmonic nanostructures in elastomers, and other fabrication techniques including laser ablation and nanocracking are summarized. Subsequently, the applications of stretchable SERS substrates in environmental monitoring, food safety inspection, biomedical analysis, and other aspects are reviewed, followed by discussions on the challenges and prospects of stretchable SERS substrates.

## 2. A Roadmap to Stretchable SERS Substrates

### 2.1. Fundamentals of the SERS Mechanism: A Brief Overview

Raman spectroscopy was discovered by the Indian physicist C.V. Raman in 1928 [20], for which he was awarded the Nobel Prize in Physics in 1930. In Raman spectroscopy, a monochromatic laser is directed onto the sample, which causes the molecules in the sample to vibrate. The laser excites the molecule to a higher vibrational energy level, and this excitation energy is then transferred to the surrounding molecules. As the excited molecules return to their original energy state, they emit photons that are scattered in all directions. Most of the scattered photons have the same energy as the incident photons, but a small fraction of the scattered photons have a different energy, which is known as Raman scattering. The energy difference between the incident and scattered photons corresponds to the vibrational energy of the molecules which is determined by the symmetricity, bonding, and crystal structure of the molecules. As a consequence, Raman scattering is physically important as a spectroscopic technique that can be used to study the structure, vibrations, and chemical composition of molecules and crystals [9,21,22]. However, the intensity of Raman spectroscopy of low-concentration samples is very limited, which inhibits its applications where high sensing sensitivity is required.

The discovery of SERS came in the late 1970s, as mentioned above, when Martin Fleischmann and Richard Van Duyne independently discovered that the Raman signal from molecules adsorbed on roughened silver surfaces was dramatically enhanced compared to Raman scattering from the same molecules in solution or on smooth metal surfaces [23]. The SERS enhancement mechanism is mainly originated from two aspects. A widely accepted mechanism is called electromagnetic field (EMF) enhancement, in which surface plasmon resonance, the collective oscillation of electrons on a metal surface in response to light, generates a localized electromagnetic field on the metal surface and leads to molecular polarization, which results in a significant increase in the Raman scattering signal [24]. The effect of EMF enhancement can be simply described by the following equation:(1)$$EF={\left|\frac{E_{loc}}{E_{0}}\right|}^{4}$$
where *EF* denotes the electromagnetic field enhancement factor, *E_loc_* denotes the local electric field excited at the hot-spot, and *E*_0_ denotes the electric field strength of the incident light, and the effect of EMF enhancement is a fourfold relation of the electric field. Another mechanism is chemical enhancement, which refers to the charge transfer or chemical bonding between the metal particles and the molecules, which alter the polarizability and vibrational frequency of the molecules, resulting in the enhancement of the Raman signal. These two mechanisms work together to make SERS an extremely sensitive analytical technique that can potentially be used to analyze trace amounts of molecules even down to individual molecules [25] in various fields such as chemistry, biology, and material sciences.

Over the years, researchers have developed a variety of SERS substrates, including roughened metal surfaces, nanoparticles, nanorods, nanoshells, and nanowires made of silver, gold, copper, and other precious metals on various substrates [26]. For instance, traditional SERS substrates are developed based on rigid matrices. However, with the development of material sciences, flexible and stretchable materials are applied to assemble SERS substrates for specific applications. Figure 2 illustrates a road map starting from the Raman spectrum to the SERS phenomenon and the development of rigid SERS substrates and flexible SERS substrates and, finally, to the emerging stretchable SERS substrates.

### 2.2. Traditional Rigid SERS Substrate

Traditional SERS substrates are usually developed based on rigid materials, such as glass, silicon, and porous alumina [27]. SERS substrates prepared with rigid materials provide uniform, stable, and reproducible SERS signals. For instance, the SERS substrate prepared by immobilizing silver nanoparticles (AgNPs) on the surface of an organosilane-functionalized glass showed good mechanical strength and chemical stability and resultant high sensing sensitivity and biocompatibility [28]. Silicon wafers were also applied to fabricate the SERS substrate for the detection of DNA, and the result shows that the Si-based SERS substrate has excellent sensitivity and specificity for multiplexed and single-based mismatched DNA detection, which is of great interest in the biological field [29].

However, traditional SERS substrates are rigid and brittle so are not feasible for some specific applications where variable plasmonic nanostructures and conformable sampling are required [30,31]. Conventional rigid substrates are usually used to detect molecular samples in liquids by dip-coating or dry samples on object surfaces by contact sampling [32]. The sampling processing of dip-coating is relatively complex and the efficiency of contact sampling is limited due to the unconformable contact with non-planar surfaces with the rigid SERS substrates. Similarly, the nanogaps providing the Raman signal are fixed, i.e., the hot-spots cannot be reconfigured once they are formed on the SERS substrates. As a result, large-size samples like viruses and cells can hardly be loaded into the hot-spots, further hindering the usability and sensitivity of traditional SERS substrates in practical applications.

### 2.3. Flexible SERS Substrate

Flexible SERS substrates are developed based on flexible materials which offer significant advantages over conventional rigid SERS substrates. For example, flexible SERS substrates can highly improve the accessibility of device preparation by cutting the samples into the right size and desired shape, and also enhance the efficiency of sampling by conformable contact scraping or wrapping the SERS substrates around irregular sample surfaces [33]. As a consequence, flexible SERS substrates have been researched to compensate for the limitations of traditional SERS substrates. Detection of pesticide residues on the surfaces of fruits [34], in situ detection of trace amount molecules [15], and label-free detection of phenol-soluble modulins [35] have been achieved by using flexible SERS substrates. In addition, studies have also shown that adjusting the spacing of surface particles can actively change the surface plasmon resonance (SPR), thus optimizing the SPR position and improving the Raman signal [15]. However, the hot-spot size of flexible SERS substrates is semi-fixed due to the limited stretchability of the substrate materials, which in turn results in unsatisfactory sampling efficiency and limited detection performance of large-size targets.

### 2.4. Stretchable SERS Substrate

Stretchable SERS substrates are fabricated by incorporating stretchable materials such as elastomers or polymers with high stretchability into SERS substrates, as mentioned above. Compared to traditional SERS substrates and flexible SERS substates, stretchable SERS substrates can not only retain their original sensing functionality even under the conditions of dynamic mechanical strain or deformation but they also show superb capability in extensively improving sampling efficiency and sensing sensitivity by controlling the layout of nanostructures on the SERS substrates. The enlarging hot-spot size would enable the high-efficiency loading of large-size samples like cells, and the recovering hot-spot size would enhance the magnitude of the electromagnetic field between the hot-spot and, thus, increase the intensity of the SERS signals [36,37]. Due to their unique properties, stretchable SERS substrates can be used for the collection and detection of various types of samples (e.g., liquids, solids, and biological samples) in a variety of settings, such as dropping solutions containing organic molecules, inorganics, and biomolecules directly onto the substrate for chemical analysis; swabbing to extract pesticide residues from irregular sample surfaces; and introducing SERS-labeled molecules into biological samples for biomedical research and clinical diagnostics. This review focuses on the fabrication and applications of the emerging stretchable SERS substrates.

## 3. Fabrication of Stretchable SERS Substrates

As mentioned above, methods for preparing stretchable SERS substrates are basically similar to those of preparing traditional and flexible SERS substrates, which involve anchoring plasmonic nanostructures such as nanospheres, nanorods, and nanostars on stretchable materials. However, the preparation of uniform plasmonic nanostructures on flexible and stretchable substrates with high stability is still challenging. Both the size, morphology of plasma nanostructures, and the properties of the stretchable matrix material are important factors that affect the performance of stretchable SERS substrates [38,39]. For example, the deciduous nanostructure of the stretchable substrates during dynamic stretching operations would highly affect the stability of the sensors [40]. As a result, the material design of the stretchable matrix and the construction of the plasmonic nanostructure on the stretchable supporting material are the key factors for the development of high-performance stretchable SERS substrates.

It is noted that most of the fabrication techniques used for preparing stretchable SERS substrates are the same as those of the traditional SERS substrates and flexible SERS substrates. As a consequence, some of the traditional nanofabrication techniques such as e-beam lithography are not presented herein. In this section, the preparation of stretchable SERS substrates is reviewed by six categories including stretchable supporting materials, followed by five typical fabrication techniques: in situ wet chemical synthesis, physical deposition, physical adsorption, and nano-embedding, as well as other unconventional methods such as laser ablation and nanocracking.

### 3.1. Stretchable Supporting Materials

Materials for the preparation of flexible SERS substrates include nanocellulose [41], cotton fabric [35], graphene/graphene oxide [42], adhesive tapes [43], and so on. All of these materials are flexible and can cover non-planar sample surfaces but with limited stretchability, which prevents their capability to control hot-spot size [44]. Stretchable supporting substrates providing the stretchability of the SERS substrates are key to the performance of the SERS device. The materials have to meet the following requirements including good mechanical stretchability for hot-spot size adjustment, high chemical stability to avoid any corrosion and dissolution from the analyte side, high physical stability to ensure the repeatability of the sensing during periodical stretching, and high biocompatibility to meet the biological application needs.

A lot of elastomers and polymers with high stretchability have been applied in the development of stretchable SERS substrates. For example, commonly used polymers including polydimethylsiloxane (PDMS) [45], polymethylmethacrylate (PMMA) [46], polyvinyl acetate (PVA) [47], and polyvinylidene fluoride (PVDF) [48] with good stretchability and flexibility have been applied for the development of high-performance flexible/stretchable SERS substrates. To further improve the sampling efficiency of flexible SERS substrates and enhance their capability for hot-spot size control, new elastomers like hydrogels are applied. These materials have been summarized in Table 1, in which the advantages and disadvantages of these materials working as supporting substrates for stretchable SERS sensors are demonstrated.

For example, polymers [49], textiles [50], silicone rubber [19], electrical tapes [51], and hydrogels [52] have been widely used to make frames for stretchable SERS substrates. PDMS has the best transparency compared to other types of polymers. High hydrophobicity and Raman enhancement factor at 150% elongation of a micro-nano-structured PDMS substrate were demonstrated by Tan et al. [53]. Polycaprolactone (PCL) membrane has excellent flexibility, biodegradability, and biocompatibility, which makes it a suitable material for tape-based stretchable sensor development. The stretchable SERS substrate prepared by decorating Ag nanowire (NW) on the tape showed that an optimized SERS signal can be obtained under the condition of 15% tensile strain [51]. Textiles have good toughness, hydrophilicity, and the porous space in the fabric structure allows rapid enrichment of analyte molecules and stabilization of SERS signals in biofluid droplet samples. A SERS substrate prepared using cotton fabrics working under the condition of 30% stretchability was reported by Garg et al. [50]. Compared to other materials, silicone rubber and hydrogels with a lower elastic modulus are popular materials for the development of SERS sensors with high stretchability. The gap distance between nanostructures can be tuned easily by applying mechanical stretching to the supporting matrices. For instance, Alexander et al. [19] prepared Au dimers on stretchable silicone rubber membranes that allowed active and reversible tuning of the interparticle gap at levels below 5 nm.

**Table 1 nanomaterials-13-02968-t001:** Stretchable material used as matrix of SERS substrates.

Stretchable Substrate	Physical Property	Advantage	Disadvantage	Ref.
Polymer elastomer	PDMS	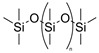	~150% elongation	Easy to obtain, low cost, good flexibility, and optical transparency (~100%)	Not resistant to high temperature	[53,54,55,56]
PCL	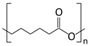	~650% elongation	Good transparency (~90%) and temperature stability (9.62%)	Insufficient mechanical strength	[57]
PMMA	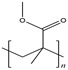	~3160 MPaelastic modulus	Excellent optical transparency (~92%) and good flexibility	Low surface hardness, easy to scratch	[58,59]
PVA	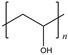	4400~5400 MPaelastic modulus	Ultrathin, flexible, stretchable, adhesive, and bio-integrable	Poor water resistance	[47]
PC	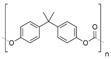	~2200 MPaelastic modulus	High tensile, flexural, and compressive strengths	Not resistant to strong alkali	[60]
	PVP	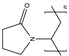	-	Biocompatible, highly plastic, and adhesive	-	[61]
Textiles	-	~30% elongation	Washable and reusable	Not resistant to high temperature	[50]
Silicone rubber	-	200~900 MPa elastic modulus	Reversible gap adjustment	Low tensile strength	[19]
Electrical tape	-	~150% elongation	Paste and peel off and cost-effective	Non-biodegradable	[51]
Hydrogels	Poly (acrylic acid)	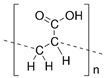	1~100 kPaelastic modulus	Stimulus responsiveness, large volume change, and biocompatibility	-	[52]

### 3.2. In Situ Wet Chemical Synthesis of Plasmonic Nanostructures

In situ wet chemical synthesis refers to the synthesis of plasmonic nanostructures on the surface of stretchable carriers through the reduction of reducing agents, thereby forming nanoparticles or continuous coatings on the carriers [62]. In situ wet chemical synthesis is widely used in SERS substrate development because it provides several merits over other methods, including being low cost and easy to scale up where high fabrication yield is required, its capability to tune the size and morphology of nanostructures by controlling the chemical reaction process, and the strong adhesion force between the substrates and the nanostructures enabling stable performance of the sensors.

A lot of flexible SERS substrates and stretchable SERS substrates have been developed based on this method [63]. For example, a low-cost, easy-to-operate, and stable PDMS-Au nanoparticles (AuNPs) composite membrane formed by dropping gold chloride (HAuCl_4_) solution directly into a PDMS microchip to construct colorimetric sensors was reported by Wu et al. [64] (Figure 3a). Gold nanostars were also grown on PDMS membranes by using the in situ synthesis method to further improve the flexibility of SERS spectroscopy [65]. In this work, AgNPs were synthesized on PDMS and worked as seeds for the growing of Au stars in the following steps. A SERS substrate constructed by Au-coated AgNPs on the PDMS surface can be obtained, as shown in Figure 3b. Chekini et al. [66] reduced metal nanoparticles on a stretchable PDMS substrate in four steps: (i) air plasma treatment, (ii) salinization of the PDMS surface, (iii) deposition of AuNP seeds, and (iv) growing of AuNPs on the PDMS surface, and found that under the condition of mechanical stretching, the particle gap increased in the stretching direction and decreased in the direction perpendicular to the PDMS stretching, resulting in color change of the sample due to the resultant plasma coupling strength change (Figure 3c). The result also indicated that nanoparticles synthesized by the in situ method can adhere to the substrate tightly even when the stretching operation is applied, and the plasma coupling as well as the electromagnetic fields between these nanoparticles can be adjusted by using the stretchable substrate.

Due to the reductive nature of 2D nanomaterials like MoS2, metal nanoparticles such as AuNPs can self-assemble directly on the MoS2 surface after reduction, which can effectively adjust the metal nanoparticle gap and improve the performance of photonic devices. As a result, many hybrid nanostructures combined with metals and low-dimensional materials have been developed based on the in situ method [67,68]. For example, three-dimensional stretchable hybrid substrates have been formed on the surface of graphene and MoS2, and high-sensitive detection of analyte molecules on arbitrary curve structures using these devices was achieved [69,70]. Li et al. [71] proposed a stretchable three-dimensional AuNPs@MoS2@GF hybrid structure and tested Raman spectra under the condition of different stretching lengths and number of stretches (Figure 3d), showing that the device is highly stable after cycles of 100% stretching and a limit of detection (LOD) as low as 10^−10^ M by testing crystal violet (CV) molecules. The authors also demonstrated that the hybrid nanostructure can be used as a cut-paste SERS substrate to cover any shaped surface.

In situ wet chemical synthesis of metal nanoparticles is a convenient and simple method for the preparation of SERS active substrates, but this method still has some limitations. Since the plasma metals are randomly arranged on the support substrate, they have poor controllability over size and shape, which in turn leads to the unsatisfactory uniformity and reproducibility of SERS active substrates. Fabrication of highly ordered nanostructures on stretchable substrates while maintaining the dynamic stability of the anchored nanoparticles on various stretchable materials with tunable functionalities is expected.

**Figure 3 nanomaterials-13-02968-f003:**
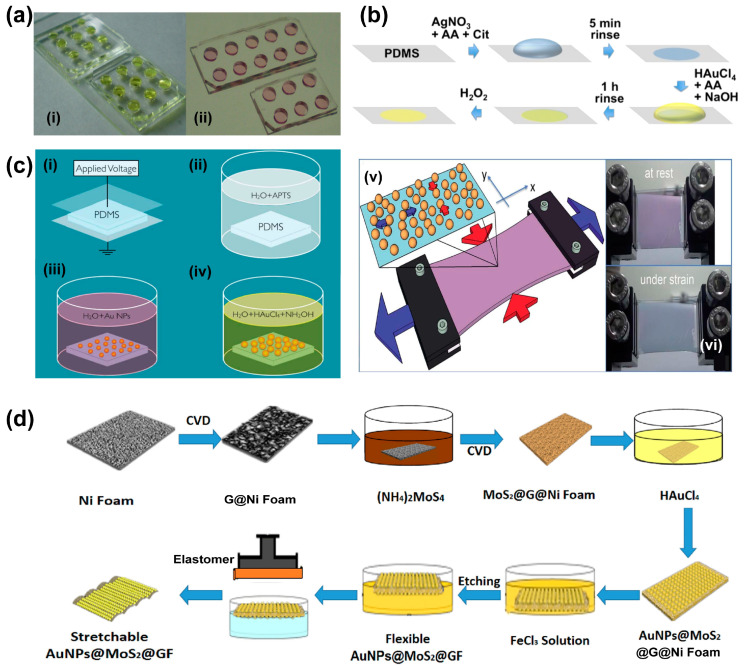
In situ wet chemical synthesis of plasmonic nanostructures. (**a**) In situ synthesis of AuNPs on a PDMS chip (**i**) with HAuCl4 solution in the reservoir and (**ii**) AuNPs grown on the PDMS surface [64]; (**b**) schematic illustration of in situ growth of AgNPs@Au-shell on a piece of PDMS [65]; (**c**) in situ growth of AgNPs on a stretchable PDMS surface in four steps, (**i**) air plasma treatment, (**ii**) salinization of the PDMS surface, (**iii**) deposition of AuNP seeds, (**iv**) growing of AuNPs on the PDMS surface; (**v**) a sketch of the experimental setup and (**vi**) an example of change of color from purple-red to blue-violet during stretching of the sample [66]; (**d**) fabrication processes of the stretchable hybrid AuNPs@MoS2@GF substrates [71].

### 3.3. Physical Deposition Method

Physical deposition technology refers to the deposition of an ordered array of nanoparticles with specific shapes on supporting substrates [72]. This method is usually achieved by physical vapor disposition technology in which condensed phase materials are transited in the vapor phase and then back to the condensed phase and deposited on the supporting substrate under the condition of high vacuum. The depositions often show excellent abrasion resistance and uniform morphology. In addition, this method allows the deposition of a variety of inorganic and organic materials with high controllability and reproducibility, which enables the development of hybrid nanostructures [73,74]. The surface formed by physical deposition is usually smooth. To enhance the plasma coupling effect, roughened or pre-patterned supporting substrates are usually applied to change the morphology of the metal film.

For example, holographic lithography or colloidal lithography [75,76] combined with selective etching [77] on substrates followed by physical deposition of Au are used to pattern uniform plasmonic nanostructures on plane surfaces. Direct coating of Au on stretchable substrates with natural rough surfaces is also a simple method to construct SERS substrates. Alternatively, the rough substrate can be obtained by nanofabrication or soft lithography using templates such as anodic alumina [78,79]. For instance, ordered inverse opal array structures composed of PDMS and gold nanoparticles (AuNPs) have been designed for uric acid (UA) detection by using a bottom-up self-assembly method. The periodic distribution of hot-spots was constructed in five steps: assembling of polystyrene (PS) monolayer, casting liquid PDMS, peeling off cured PDMS, sputtering of Au, and transfer of AuNPs film, as sketched in Figure 4a [80].

Interestingly, bioinspired SERS substrates have also been developed for high-performance sensing, in which natural biological materials such as taro leaves [81], reed leaves [49], and cicada wings [82,83] were used as the replication templates. Figure 4b illustrates the procedure of fabricating micro and nanostructured SERS sensors using a natural reed leaf as the template [49]. The substrate can be obtained simply in three steps: (i) PDMS template duplication from a reed leaf by soft lithography, (ii) PDMS reed leaf duplication from the PDMS template, and (iii) deposition of Au on the PDMS reed leaf. The detection of metabolites in sweat by using the SERS substrate was achieved.

However, most of the substrates prepared by the physical deposition method are prepared on a 2D plane system, which leads to a limited number of hot-spots on the substrates. Recently, Kumar et al. [84] prepared a 3D SERS substrate by using oblique angle deposition (OAD) technology to deposit long silver nanorods (AgNR) onto a PDMS film with a pre-strain of 30%, and bacterial samples were directly added to the substrate (Figure 4c). SERS measurements were carried out on SERS substrates under tensile and released conditions, respectively. The results showed that the Raman signal intensity of the substrate under the condition of the released state was ~10 times higher than that of the pre-stretched condition (Figure 4d). The superb performance of the SERS substrate is due to the 3D cage structure of the nanowrinkled surface with better bacteria and metal nanostructures surface contacts and, thus, more hot-spots. Using nanowrinkling for enhancing the SERS signal was also reported when using shape memory polystyrene substrate [85]. This strategy was also reported by Singh et al. [86], in which silver nanorods were deposited on flexible PDMS and polyethylene terephthalate (PET) materials by OAD technology and a silver nanocolumnar film diagram with an inclination angle of >75° on the surfaces was achieved. In situ SERS measurements of this flexible substrate under tensile strain revealed that after 100 cycles of tensile testing, the nanorods remained undamaged, indicating that the nanostructure fabricated by this method is pretty durable (Figure 4e).

Inorganic materials such as graphene are ideal sources for SERS substrate development [87,88]. Studies have shown that graphene fold structures can improve the performance of hybrid SERS substrates [89,90]. For example, Chen et al. [40] prepared a high-performance hybrid SERS substrate by depositing Au nanoparticles onto a pleated graphene surface (Figure 4f). The height and distribution of the folds in pleated graphene can be tuned by changing the ethanol conditions. The results show that AuNPs are evenly distributed on the pleated graphene surface, and the hybrid substrate is capable of detecting R6G molecules at ultra-low concentrations of 10^−9^ M with an enhancement factor (EF) as high as ≈1.19 × 10^7^. The substrate also shows high tensile properties, which can be applied to any curvature surface and can detect multiple analytes at the same time, making it ideal for detecting water contaminants and analyzing crop pesticide residues.

**Figure 4 nanomaterials-13-02968-f004:**
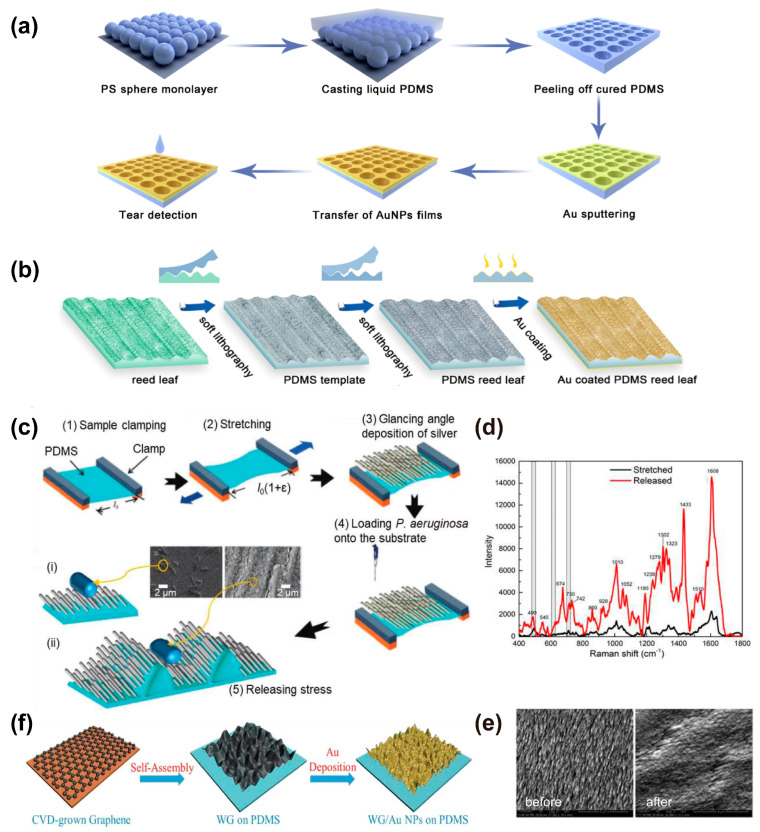
Physical deposition of SERS substrates. (**a**) A schematic illustration of SERS substrate preparation by physical deposition method and the detection of uric acid in tears [80]; (**b**) a schematic illustration of plasmonic nanostructure fabrication using a natural reed leaf as the template [49]; (**c**) a schematic illustration of the fabrication of the AgNR−PDMS substrate and schematic showing *P. aeruginosa* on (**i**) pre−stretched and (**ii**) wrinkled AgNR−PDMS substrates [84,91]; (**d**) SERS spectra of *P. aeruginosa* on pre−stretched and released (wrinkled) AgNR−PDMS substrates; (**e**) SEM images of a flexible AgNR−PDMS substrate before and after one hundred tensile loading cycles [86]; (**f**) a schematic illustration of the preparation of WG/AuNPs hybrid SERS platform [40].

SERS substrates prepared by physical deposition technology have good uniformity and reproducibility, offering a powerful tool for creating both flexible and stretchable SERS active substrates. However, this technique still has some limitations. For instance, NPs are directly deposited and assembled onto the surfaces of stretchable polymers. When the substrates are stretched, NPs can easily fall off under the action of tension, resulting in poor stability of the SERS substrates. In addition, physical deposition usually requires special instruments and high vacuum conditions, and special techniques like the above-mentioned OAD are required to obtain patterns with special designs and morphology.

### 3.4. Physical Adsorption Method

Physical adsorption is the simplest method to assemble SERS substrates, in which ready-to-use nanomaterials are immobilized onto supporting substrate surfaces through weak bonds such as electrostatic attractions, hydrogen bonding forces, van der Waals forces, or hydrophobic interactions. Before the adsorption operation, the surfaces of the supporting substrates are usually treated by physical or chemical methods, such as changing the surface charge [26] and the hydrophilicity [92] to adjust these interactions. After the modification, ready-to-use plasmonic materials can be transferred onto the substrate, for example, by dip-coating [51], to assemble SERS substrates. The key advantage of the physical adsorption method is that it is easy to operate and cost effective compared to the other fabrication methods.

Silver nanowire network film (AgNWNF) has been deposited on PDMS as a substrate to assemble stretchable SERS substrates [26]. To firmly adsorb the nanowires, the PDMS substrate surface was treated with ultraviolet ozone (UVO) to generate hydroxyl groups, and the UVO treatment can oxidize and remove grease and organic matter from the substrate surface to achieve a cleaning effect. The excess hydroxyl group can strongly adsorb the deposited AgNWNF on the PDMS surface (Figure 5a). The adhesion strength tests showed that the nanomaterials deposited on the UVO-treated PDMS have better adhesion and stability properties compared to those deposited on the one without UVO treatment. To further enhance the adhesion force between the nanomaterials and the substrates, chemical methods have also been widely applied. The formation of covalent bonds through chemical reactions is often reported on rigid substrates, and many works have also been conducted on flexible or stretchable substrates. Mir-Simon et al. [93] reported the adsorption of AuNPs to the 3-mercaptoproptrimethoxysilane-functionalized PDMS substrate through Au-S chemical bonds and found that the AuNPs array was evenly distributed on the PDMS substrate (Figure 5b). The modification procedures are as follows: the first step is an activation of the PDMS with H_2_O_2_ and to salinize the PDMS surface with HCl solution, and then Au nanoparticles were covalently bound to the thiol-terminated PDMS through the salinization groups and Au-S bonds. A tensile stress of 60% was applied to the sample, and the plasmon wavelength decreased linearly from 664 nm to 591 nm, which indicates that the AuNP array fabricated by this method has high stability and uniformity [93].

To further narrow the gap between metal nanoparticles and prepare high-density nanoparticle structures on SERS substrates, a tightly and uniformly arranged AuNP monolayer was autonomously assembled at the water/oil interface and then transferred to a stretchable PDMS substrate by Langmuir–Blodgett transfer technology [94], as shown in Figure 5c [95]. This method is very simple and can control the hot-spot size precisely. It is reported that with the increase in the diameter of AuNPs, the average gap size can easily reach below 1 nm. Nanostructure arrays preparing at the organic/water interface were also reported, and by removing the upper organic solution and adding the mixture of PDM on the interface, a stretchable substrate with nanostructure arrays on the cured PDMS surface can be obtained, as shown in Figure 5d [96]. By combining the high-density gold nanoparticle monolayer with the stretchable PDMS substrate, the stability, reproducibility, and sensitivity of the SERS substrate can be significantly improved [96].

**Figure 5 nanomaterials-13-02968-f005:**
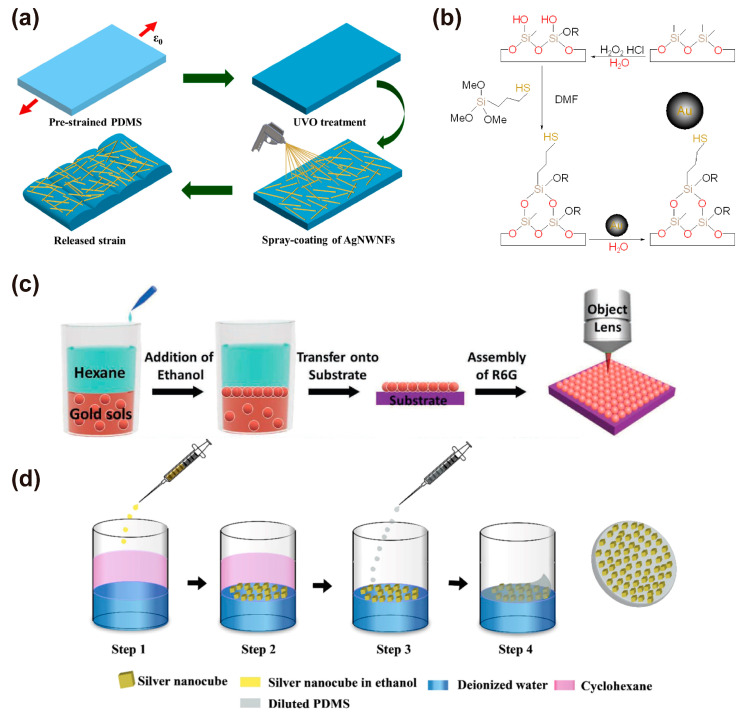
Plasmonic nanostructure fabricated by physical adsorption method. (**a**) Schematic illustration of fabricating AgNWNFs-deposited SERS platform on PDMS nanowrinkles [26]; (**b**) a scheme of chemical approach to synthesize the derivative substrate [93]; (**c**) a scheme for fabricating and transferring Au nanoparticle monolayers from the water/hexane interface and an illustration of the SERS experiment [95]; (**d**) a schematic illustration of as-proposed fabrication process of AgNCs@PDMS SERS substrate [96].

There are many works related to the physical deposition method; herein, we just listed a few of them. However, immobilization through physical adsorption usually suffers from drawbacks such as desorption of the nanostructures. It is expected that the adsorption of nanostructures can be further improved by pretreatment of the support substrate, which is conducive to improving the stability of SERS detection. Also, it is promising to develop more nanomaterials such as the hybrid nanomaterials to expand the functionality and sensitivity of the SERS sensors.

### 3.5. Embedding of Nanomaterials

Nanomaterials with high surface free energy are easy to agglomerate and oxidize and have poor stability in a free environment. The embedding of plasmonic nanomaterials into host materials such as PDMS can effectively inhibit aggregation of the nanomaterials and also create a new type of SERS substrate with enhanced optical properties. The typical embedding method is shown in Figure 6a, in which nanomaterials are adsorbed onto a silica wafer, and a liquid polymer precursor is poured on the solid surface and embeds the particles. After the polymer cures, the nanomaterial–polymer composition is peeled off from the substrate and a transparent and flexible SERS sensor is obtained [97]. The optical transparency property of the substrate allows light interaction with the underlying contact surface and the detection of analytes adsorbed on arbitrary surfaces. The sensor has achieved the detection of a trace amount of benzenethiol of 10^−8^ M and an EF of ~1.9 × 10^8^. In addition, the SERS sensors can maintain high sensitivity even after 100 cycles of stretching, bending, and torsion deformations (Figure 6b).

Most current research has focused on spherical nanostructures, and the tunability of plasma properties of stretchable substrates still needs to be further explored [98]. In the work reported by Wen et al. [55], silver-plated gold nanorods were embedded in PDMS to investigate the adjustability of the plasma resonance of the substrate. The preparation process is almost the same as that demonstrated in Figure 6a. First, gold-plated gold nanorods with a silver shell are deposited on a silicon surface, and then the mixture of PDMS is poured onto the silicon and peeled off after the PDMS is cured. Finally, a PDMS device with silver-coated gold nanorods embedded is successfully obtained. Compared with spherical nanorods, silver-coated gold nanorods usually exhibit synthetic tunable plasma resonance over a wider wavelength range and have stronger electric field enhancement, so that the resulting substrate has a high degree of stretchability and stability [55].

The above-mentioned silver-plated gold nanorods are randomly embedded in the elastomer, and the hot-spot structure between the gold nanorods cannot be continuously adjusted. Yan [61] reports an elastic SERS substrate that can simultaneously and continuously tune a large number of AuNP nano-assemblies embedded in the substrate by reversible deformation of the elastic substrate. As shown in Figure 6c, AuNPs are self-assembled into linear nanochains (AuNCs) in an aqueous medium by suitable thiol or phosphine ligands, and then polyvinylpyrrolidone (PVP) powder is added to the AuNCs solution. The obtained solution is dropwise cast on the PDMS substrate for drying, and finally, the blue film is peeled off from the PDMS substrate to form an AuNCs-PVP film, whereby AuNCs are embedded in the PVP matrix. The film is a new type of SERS substrate and the EF can be mechanically modulated in the range of 10^0^ to 6.8 × 10^7^, which can be used to directly analyze small molecular analytes in complex biological samples and foods.

Nanomaterials are combined with stretchable matrices by embedding, and the surface of the SERS substrate formed is smooth, which makes it easy to cover the sample surface and achieve perfect contact sampling. The unique surface property would also avoid damage to sample surfaces, resulting in higher and more uniform Raman enhancement on the entire surface. With the development of stretchable matrices, a large number of plasmonic nanomaterials can be embedded in the stretchable matrices, which can facilitate the simultaneous and continuous tuning of the SERS enhancement performance via simple mechanical operations.

### 3.6. Other Methods

In addition to the above-mentioned fabrication techniques, there are also many unconventional methods by which plasmonic nanostructures can be obtained on stretchable substrates. For example, the laser ablation method has been used to create SERS substrates in an easy, fast, and low-cost manner, in which metal nanoparticles like Ag and Cu can be readily reduced under the condition of high temperature as a laser beam is focused on the sample surface [99,100]. Laser-scribed graphene (LSG) technology has been used to develop flexible SERS substrate by generating silver dendrites (AgD) directly on a soft substrate, in which the laser allowed the controlled reduction of graphene oxide for the following growing of AgD by electrodeposition. The SERS signal can be enhanced by the combination of the inherent chemical enhancement of graphene and the electromagnetic enhancement of the Ag nanostructures [90]. Figure 7a shows images captured during the fabrication process: (i) a sample of flexible LSG and (ii) GO/LSG, (iii) a zoomed-out view of the LSG surface, and (iv) a zoomed-in view of AgD [90]. Similarly, a SERS substrate fabricated by femtosecond laser irradiation combined with Au coating and annealing was also reported by Cao et al., and an EF as high as 8.3 × 10^7^ was achieved [101]. It is easy to achieve large-scale fabrication using the laser ablation method; however, tuning the size and morphology by this method is still challenging.

Breaking metal nanofilm, also called “nanocracking”, is also an efficient way to generate plasmonic nanostructures. For example, silver nanopaste (AgNPA) was employed to prepare an ultrasensitive wafer-scale SERS substrate [103]. The fabrication process is briefly introduced as follows: AgNPA was spin-coated on Si wafers, and high-density Ag nanocracks with small gaps can be spontaneously generated after the evaporation of solvent. The small gaps can provide abundant hot-spots for the ultrasensitive detection of analytes down to single molecules [103]. The nanocracking method is a lithography-free method which can also be triggered by machinal stretching. Li et al. reported a large-area pattering of plasmonic nanostructures on ultrathin flexible thermal plastic films through the deposition of Au films, followed by a mechanical stretching process, as shown in Figure 7b. The inset images in Figure 7b illustrate the morphology of the nanocracks under different strains applied. Both the size and arrangement of the nanopatterns can be well controlled by varying the thickness of the nanofilm and the mechanical stretching [60]. The authors also demonstrated the transfer of the ultra-flexible sensing films onto curved surfaces for conformable sensing.

Another strategy to fabricate high-performance SERS substrates is to combine the listed methods. For example, one can prepare plasmonic nanostructures by in situ synthesis method and then deposit another material onto the nanostructure by a physical deposition method or adsorption process to enhance the functionalities of the substrates. Many devices have been developed based on the hybrid method. For example, a facial method to fabricate a flexible and transparent SERS substrate based on PDMS film modified with a Ag/Au nanowire (NW) forest was reported [102]. Au seeds were anchored on the PDMS surface and an in situ chemical synthesis method was applied to grow Au nanowires on the PDMS surface; thereafter, magnetron sputtering coating of Ag nanoparticles was conducted to modify the vertically-aligned Au NWs with Ag nanoparticles (Figure 7c) [15]. The Ag/AuNWs/PDMS flexible substrate shows good toughness and a remarkably improved EF value of 6.74 × 10^6^ of the Ag/Au NWs/PDMS film was achieved by mechanical stretch. The stretched film is also suitable for the in situ detection of pesticides on the surface of crops, with the advantages of low cost and easy preparation [102]. One can believe that the compensation of the fabrication techniques would highly improve the performance of the new-coming SERS substrates.

## 4. Applications of Stretchable SERS Substrates

Stretchable SERS substrates have been developed for a variety of cutting-edge applications, including environmental analysis, food safety and pesticide detection, medical analysis, public safety, and other fields. Here, we briefly describe specific examples of stretchable SERS substrates in the aforementioned fields and both the limitations and the potential of stretchable SERS substrates in these applications are discussed.

### 4.1. Environmental Analyses

With the rapid development of global industrialization, the problem of environmental pollution is becoming more and more prominent, posing a threat to human existence and the ecological environment, especially from molecular pollutants. Therefore, the development of high-performance SERS substrates for real-time and on-site monitoring of pollutants is urgently needed. Among the current environmental pollution problems, organic pollutants are persistent, difficult to degrade, and highly toxic, posing a significant hazard to the environment. As a result, a lot of stretchable SERS substrates targeting these pollutants have been developed.

For example, a stretchable and flexible SERS substrate developed for the detection of organic dyes was reported by Tan et al. [53] The results show that after stretching and shrinking, the PDMS substrate exhibits an irregular wrinkled structure with abundant gaps and grooves that function as hot-spots, thereby improving the testing sensitivity. A detection limit of 1 × 10^−7^ M was achieved by using the substrate [53]. Benzenethiol (BT), a highly toxic organic pollutant, can contaminate groundwater, watercourses, or sewage systems [104]. Kang et al. [18] prepared a stretchable SERS substrate by colloidal lithography followed by electrostatic assembly of gold nanoparticles on the caps. The substrate can be tuned reversibly under external strain and provides a promising mechanism for optimizing the SERS sensitivity (Figure 8a). BT concentrations as low as 10 nM can be detected by using this substrate. The results also show that when detecting BT molecules, the stretchable SERS substrate under the condition of 20% strain is 1000 times more sensitive than that obtained at zero strain (Figure 8b), and the EFs of the sensor under the condition of strains are much higher than those without strain (Figure 8c).

Malachite green (MG) is mainly used in aquaculture and industrial production, and its residue is often found in water sources and aquatic products, causing serious harm to human health [105]. Based on the stretchable SERS concept, Kumar et al. [81] reported a stretchable biomimetic SERS substrate for the detection of MG (Figure 8d). The substrate was prepared by using taro leaf with microcavities as the template for PDMS replication, followed by the plating of silver nanostructure on the PDMS surface. The Ag-coated microcavity-structured PDMS substrate exhibits high adhesion and hydrophobicity properties, showing a “rose petal effect”, which can highly enhance the SERS intensity. A significantly high enhancement factor of ~2.06 × 10^5^ and a detection limit of ~10^−11^ M for MG with high reproducibility were obtained [81]. The applications of stretchable and flexible SERS sensors for monitoring environmental pollutants are still ongoing, and many of them have been extended to fields such as food safety monitoring.

### 4.2. Food Safety Monitoring

Pesticides, food additives, and other food pollutants can cause food poisoning within a short period and may also accumulate in the human body for a long time. This can lead to fatal harm to humans including but not limited to disrupting body metabolism, damaging genes, and causing cancers [106,107,108]. SERS technology has proven to be an effective method for the quantitative or qualitative testing of pesticides and other hazards on fruits and vegetables, but the in situ sampling from curved or arbitrary geometry surfaces of fruits and vegetables by traditional rigid SERS substrates is inefficient. Stretchable SERS substrate has better conformable contact with samples and higher loading efficiency than rigid sensors and, thus, higher sensitivity and useability, which can better meet the requirements of food safety analysis. Here, selective application examples of stretchable SERS substrates in the analysis of food contaminants are presented.

Pesticides are the most common source of contamination in food and tend to remain on the surface of vegetables and fruits; they can be effectively detected by stretchable SERS substrates via conformable sampling [109]. For example, a stretchable SERS substrate for the detection of carbendazim (CBZ) was reported, in which polymorphic silver nanoparticles modified with zinc oxide nanorods were decorated on stretchable substrates by the soft lithography method [109]. The 3D hybrid substrate has a high-roughness nanostructure that can enhance the signals of CBZ and achieve a detection limit down to 10^−3^ ppm. Similarly, Ma et al. [102] prepared a PDMS membrane-based SERS substrate modified with Ag/Au nanowire forests for trace pesticide detection under various conditions. The SERS activity of the stretchable device can be effectively enhanced by mechanical stretching with an enhancement factor of up to 6.74 × 10^6^. A detection limit of 10^−6^ mg/mL for parathion (MP) on tomato surfaces proves that this device is suitable for rapid, on-site detection of pesticides on non-planar surfaces of vegetables and fruits (Figure 9a). Li et al. [60] demonstrate the extension of controlled lithography-free nanofabrication methods for the construction of ultrathin flexible photonic surfaces that can be transferred to curved surfaces of arbitrary topology to form conformal in situ SERS sensors. Thiram residue on apple skin with a detection limit as low as 0.48 ng/cm^2^ was realized by using this nanopatterned membrane, as shown in Figure 9b. Liang et al. [110] reported the preparation of Ag-SiO_2_ composite nanoparticle structures on stretchable PDMS substrates by co-deposition technique and the detection of thiram on various fruit surfaces. The device showed excellent detection limits of 10^−13^ M (Figure 9c). A sandwich-based 3D SERS sensor G@AgNPs@G/PDMS for in situ and label-free detection of malachite green (MG) was reported by Fan et al. [111]. The PDMS films can be easily shrunk for 3D structure construction, thus providing advantages in terms of enhancement capabilities and light-matter interactions. Meanwhile, the G@AgNPs@G/PDMS film has a strong adsorption capacity to extract molecules from food surfaces. Detection of MG on fish surfaces as low as 10^−13^ M was presented. Another stretchable SERS substrate developed based on a biocompatible poly(ε-caprolactone) membrane embedded with a 3D periodic wavy microstrip array and an array of nanoslots along the stretching direction was also used to detect MG [57]. The stretched polymer surface with plasmon resonance nanostructures can produce more than 10-fold signal enhancement compared to the unstretched membranes. The results show that in situ detection of MG molecules as low as 0.1 μM on irregular surfaces of green mussels by using the Ag-deposited stretched substrate can be achieved (Figure 9d).

Food additives are chemically synthesized or natural substances that can be added to food to improve the color, aroma, and taste of the food, or to achieve certain preservative effects [113,114,115]. However, the misuse of food additives can adversely affect human health. Kitahama et al. [47] reported an ultrathin, stretchable, and adhesive “Place&Play SERS” substrate, which was fabricated by deposition of gold on polyvinyl alcohol (PVA) nanogrid substrate for the detection of the biocide methylchloroisothiazolinone (CMIT) on orange peels (Figure 9e). The gold/PVA nanonet substrate showed high SERS signal enhancement of ~10^8^ which enabled the detection of ~10 nM R6G with high reproducibility. Barveen et al. [112] synthesized gold nanostars (AuNSs) directly on polymethylmethacrylate (PMMA) membranes for surface-enhanced Raman spectroscopy (SERS) applications. The transparent and flexible AuNSs/PMMA SERS substrate allowed for the real-time in situ detection of ciprofloxacin (CPX) and chloramphenicol (CAP) on chicken wing samples through the use of a fiber-coupled Raman probe (Figure 9f). The membrane substrate exhibits high sensitivity, high enhancement factor (2.03 × 10^9^), low detection limit (3.41 × 10^−11^ M), and superior multi-detection capabilities including good homogeneity and reproducibility (<7.32%). Tang et al. [116] constructed a flexible dual-plasma SERS substrate for the detection of residuals of triclosan in food samples. Due to the enhanced electromagnetic field from the plasma coupling of the AuNP arrays and aligned AgNWs, the substrate exhibits good sensitivity and reproducibility in a variety of situations, including 100 cycles of stretching, bending, and twisting, with a detection limit for melamine as low as 10^−8^ M. From the above examples, one can see that stretchable SERS substrates have been powerful tools in monitoring the misuse of food additives or illegal additions, especially where non-destructive, real-time, and in situ detection is needed.

### 4.3. Biomedical Applications

In recent years, SERS has gradually become a promising sensing technology in the fields of biomedical research, and many stretchable SERS sensors have been developed to meet the increasing needs in these fields such as healthcare monitoring and diagnosis of diseases.

For example, pseudomonas aeruginosa is a conditionally pathogenic bacterium that prefers moist environments and colonizes infection-prone injury sites, making it one of the major pathogens in hospital-acquired infections. Traditional techniques for identifying infectious agents often require prior cultivation in appropriate media, which is time consuming. Stretchable SERS plays an important role in pathogen detection, especially where high-sensitive detection is needed. For example, a stretchable SERS substrate prepared using silver nanorod arrays on PDMS showed that the bent substrate is much more sensitive than the regular substrate when detecting the same concentration of bacterial suspension [84].

In addition to detecting pathogens, stretchable SERS substrates are also used to detect biomarkers for clinical diagnostics, where usually molecular biomarkers in human body fluids are detected [117]. Al et al. [61] embedded plasmonic gold nanoclusters (AuNC) in polyvinylpyrrolidone (PVP) substrates for the detection of hypoxanthine in serum. The tunability of the enhancement factor in the range of 100~6.8 × 10^7^ was achieved by adjusting the nanostructure and the properties of the substrate array through reversible mechanical deformation. At the same time, the PVP substrate allows diffusion of small molecule markers into the hot-spots, which in turn excludes interference from other components. Liu et al. [118] demonstrated a highly scalable, low-cost, ultrathin, stretchable, wearable SERS sensor fabricated based on gold nanonet (Figure 10a). The sensor can be fabricated into any shape and worn on virtually any surface for label-free, large-scale in situ sensing of a wide range of analytes on virtually any arbitrary surface. The wearable SERS sensors have demonstrated excellent utility in detecting human sweat biomarkers. A multifunctional wearable e-skin with hierarchical micro-nanostructures for the detection of metabolites in human sweat was also reported, as shown in the schematic in Figure 10b [49]. The Au-coated PDMS reed leaves are thin, pliable, and soft, making them suitable as a wearable sensor for lactic acid and fatty acids detection from sweat samples [49]. A novel erasable and reproducible plasmonic hot-spot developed by synthesizing silver nanostructures on liquid metal was reported recently, and the hot-spots can be removed in alkaline solution. The self-adhesive, biocompatible, and stretchable device is able to detect glucose concentrations as low as 1 ng/L [119].

Stretchable SERS substrates can also integrate with other technologies such as paper-based microfluidics to achieve desired functions. For example, a paper-based wearable sensor for continuous and simultaneous quantitative analysis of sweat loss, perspiration rate, and metabolites in sweat was developed by Mogera et al. [120], in which a serpentine paper layer provides mechanical strain isolation, making the device soft, flexible, and stretchable (Figure 10c). The sensor is capable of sensitively detecting and quantifying uric acid (UA) in sweat at concentrations as low as 1 μM [120]. Similarly, Chowdhury et al. [56] constructed heart-shaped nanoparticle dimer arrays on PDMS substrates with high aspect ratios, which can maintain reliable SERS performance even under bending (up to 100° angle) or stretching (up to 50% stretching) conditions. The device is promising for applications in the field of personalized medicine and remote patient monitoring (Figure 10d). In addition, a SERS substrate for the detection of hydrophobic biomolecules developed by depositing conjugated molecules and Au particles on PDMS was reported by Wang et al. [121]. The substrate showed acceptable efficiency due to the tip-focusing effect and high durability even after 100-cycle bending by testing cholecalciferol at ultra-low concentrations (10^−10^ M).

### 4.4. Other Applications

In addition to environmental analysis, food safety monitoring, and biomedical uses, stretchable SERS substrates have also been widely used for public safety inspections. Illicit drugs pose a serious threat to social security and physical health, such as cocaine, a common drug, which can affect the brain and other vital organs when ingested in large quantities and seriously jeopardize lives [122,123]. A stretchable SERS substrate with high sensitivity is an excellent candidate for the tests. For instance, Zhang et al. reported a wearable glove-based SERS for rapid sampling and on-site detection of trace amounts of tramadol and midazolam at concentrations of 69.19 ng/mL and 35.03 ng/mL, respectively, as well as for the accurate identification of methamphetamine in complex binary mixtures [124]. The efficient analytes extraction was achieved by the design of the substrate with flexible adhesive tape, as shown in the schematic diagram demonstrated in Figure 11a [124]. Maddipatla et al. [125] developed a novel pleated-structure-based SERS substrate for the detection of cocaine. The substrate was developed by depositing silver nanoparticle (AgNP) ink onto a pre-stretched thermoplastic polyurethane (TPU) substrate using gravure printing, followed by the release of the TPU (Figure 11b). Dense hot-spots were formed on the pleated structure, and the detection of cocaine under the condition of various stretches was conducted as shown in Figure 11c [125].

The stretchable property of the SERS substrate is also suitable for applications in the biological field. As discussed above, large-size (compared to molecules) cells and proteins can be loaded in hot-spots with high efficiency via elastic stretching and achieve high sensing sensitivity by releasing the stretching. For example, a SERS substrate with active hot-spots constructed on hydrogels for detecting *cytochrome* and crystal violet was reported by Mitomo et al. [126]. The upper sketch in Figure 11d shows the preparation of the tunable plasmonic nanostructure by the self-assembling of AuNPs on glass, followed by the transfer of AuNPs onto a piece of gel membrane, and the lower sketch in Figure 11d demonstrates the working procedure of detecting target molecules by contraction of AuNPs. Figure 11e,f present Raman signals of *cytochrome c* and crystal violet under three conditions: (a) “open form”, where analytes were injected into a gel in an expanded state, (b) “closed form”, where analytes were injected into a gel in a contracted state, and (c) active gap control as “open-to-closed form”, where target molecules were injected into an expanded state gel and analyzed after contraction. The results indicate that active control of stretchable SERS substates can highly enhance the sensing sensitivity.

Stretchable SERS substrates have been proven to be a powerful tool for the identification of molecules. The applications of SERS substrates can be extended to many fields in the future, including but not limited to the detection of trace amounts of explosive substances and illicit drugs in safety inspections and also biomolecule detection in biology research.

## 5. Conclusions and Remarks

Stretchable SERS substrates have attracted significant interest in recent years due to their outstanding performance and potential applications in various fields such as environmental monitoring, food safety inspection, and biomedical analysis. In this review, the recent development and applications of the stretchable SERS substrates are summarized, and a roadmap for the development of SERS substrates is introduced. Fabrication techniques for stretchable SERS substrates and the cutting-edge applications of these stretchable SERS-active substrates are introduced. This review provides an overview of the development of SERS substrates and sheds light on the design, fabrication, and application of novel, highly efficient stretchable SERS systems, especially for future applications where tunable hot-spots and Raman spectra are needed.

The development of stretchable SERS substrates is still in its early stages, and many challenges need to be addressed, such as reproducibility, scalability, and cost. Addressing these challenges can lead to the development of more efficient and practical stretchable SERS substrates. The continued exploration of new stretchable supporting material, surface functionalization strategies, and nanofabrication techniques can also lead to discoveries and advancements of new stretchable SERS substrates. For example, the mechanical properties of the substrate must be optimized to ensure that the substrates can be stretched without breaking or losing their SERS activity; the surface functionalization methods should be improved to further enhance the adhesion force between plasmonic nanostructures and the stretchable substrates to avoid any deciduous behaviors; and the plasmonic properties of the nanostructure should also be promoted by optimizing the design of the nanomaterial composite and the shape and size of the nanostructure. Additionally, it is also important to create new SERS substrates with advancements such as renewability, biodegradability, and reducibility to fulfill the needs of applications.

## Figures and Tables

**Figure 1 nanomaterials-13-02968-f001:**
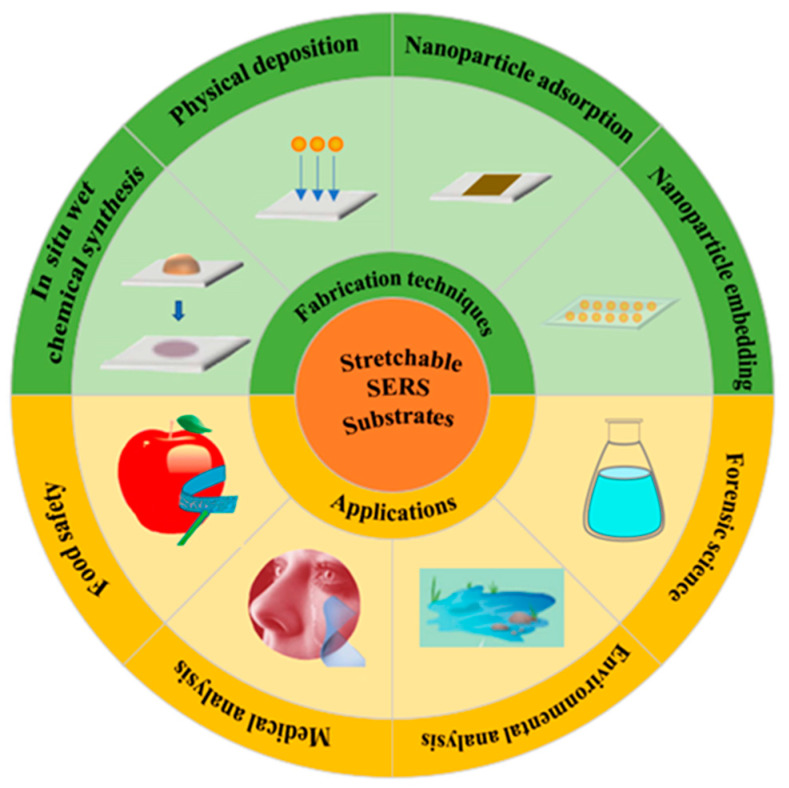
Fabrication and applications of stretchable SERS substrates.

**Figure 2 nanomaterials-13-02968-f002:**
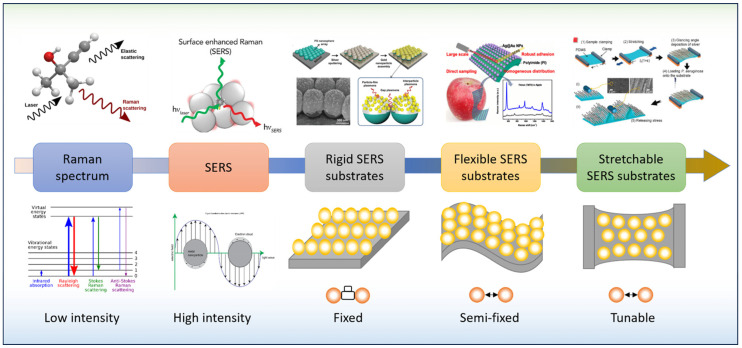
Evolution of stretchable SERS substrates.

**Figure 6 nanomaterials-13-02968-f006:**
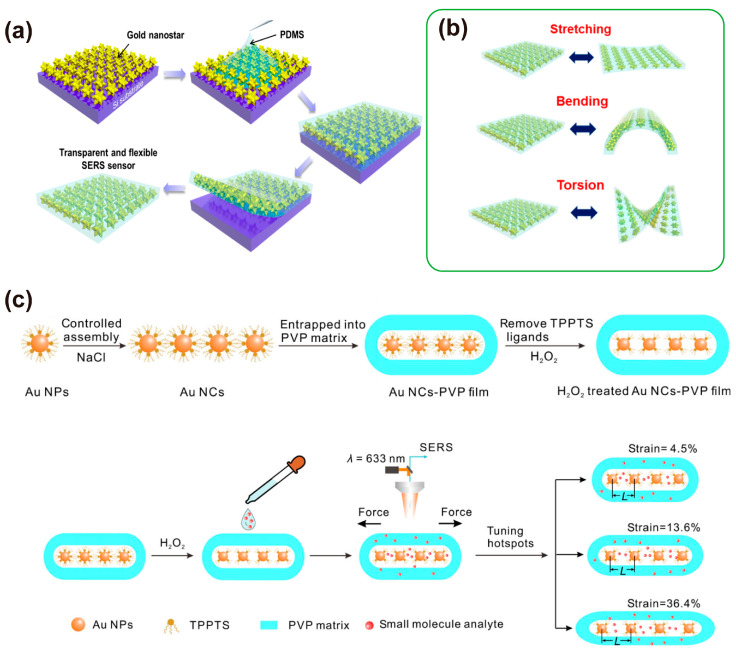
Fabrication of SERS substrates by embedding nanomaterials in elastomers. (**a**) A general schematic illustration of the fabrication of flexible SERS sensor by embedding nanomaterials in elastomers and (**b**) the typical working statutes of the stretchable SERS substrate [97]; (**c**) a schematic of the preparation procedure of AuNCs-PVP film and the detection of small molecule analyte [61].

**Figure 7 nanomaterials-13-02968-f007:**
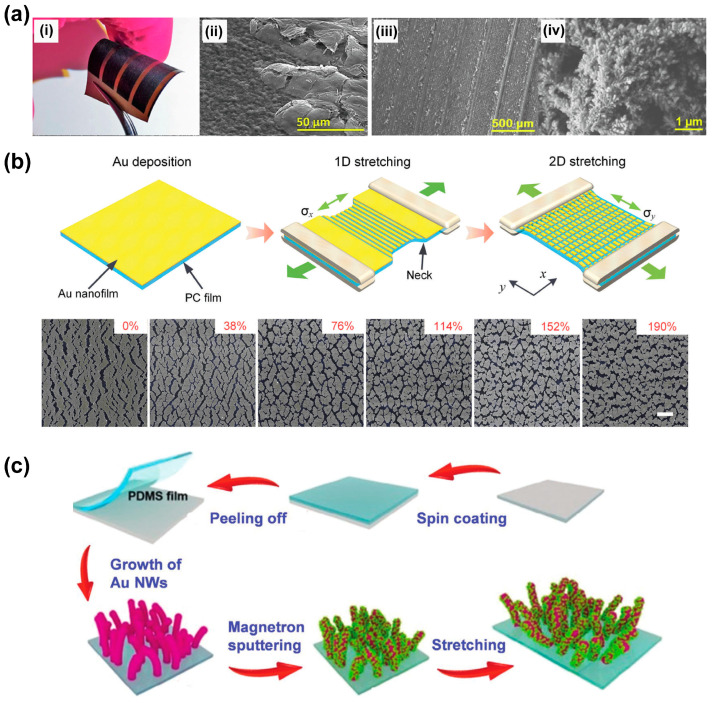
Other methods for stretchable SERS substrate fabrication. (**a**) Laser ablation method and example images captured during the fabrication process: (**i**) sample of flexible LSG, (**ii**) GO/LSG, (**iii**) zoomed-out view of the LSG surface, (**iv**) zoomed-in view of AgD; (**b**) a schematic illustration of the fabrication of the ultra-flexible SERS substrate by nanocracking and the images of the nanocracks [90]; (**c**) a schematic illustration of the hybrid methods for the preparation of stretched Ag/Au NWs/PDMS film [102].

**Figure 8 nanomaterials-13-02968-f008:**
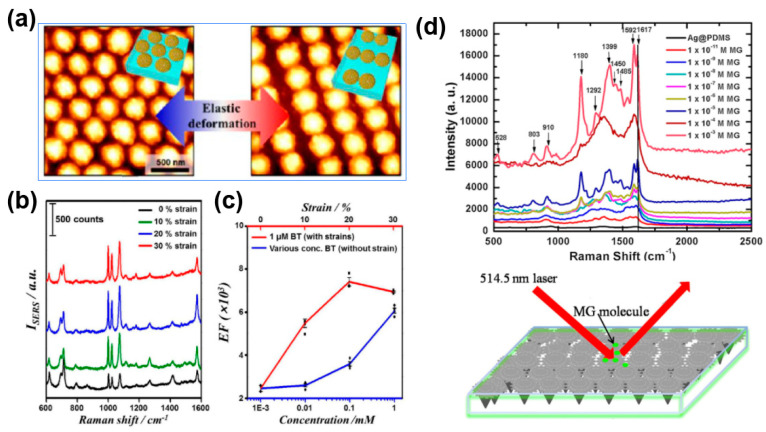
Stretchable SERS sensors for environmental analysis. (**a**) AFM images of tunable SERS substrate surface under the condition of stretching; (**b**) Raman spectra of 1 μM BT−adsorbed plasmonic cap arrays as a function of the applied strain and (**c**) comparison of the enhancement factor [18]; (**d**) schematic of detecting MG by using the micro-cavity array PDMS SERS substrate and the Raman signals [81].

**Figure 9 nanomaterials-13-02968-f009:**
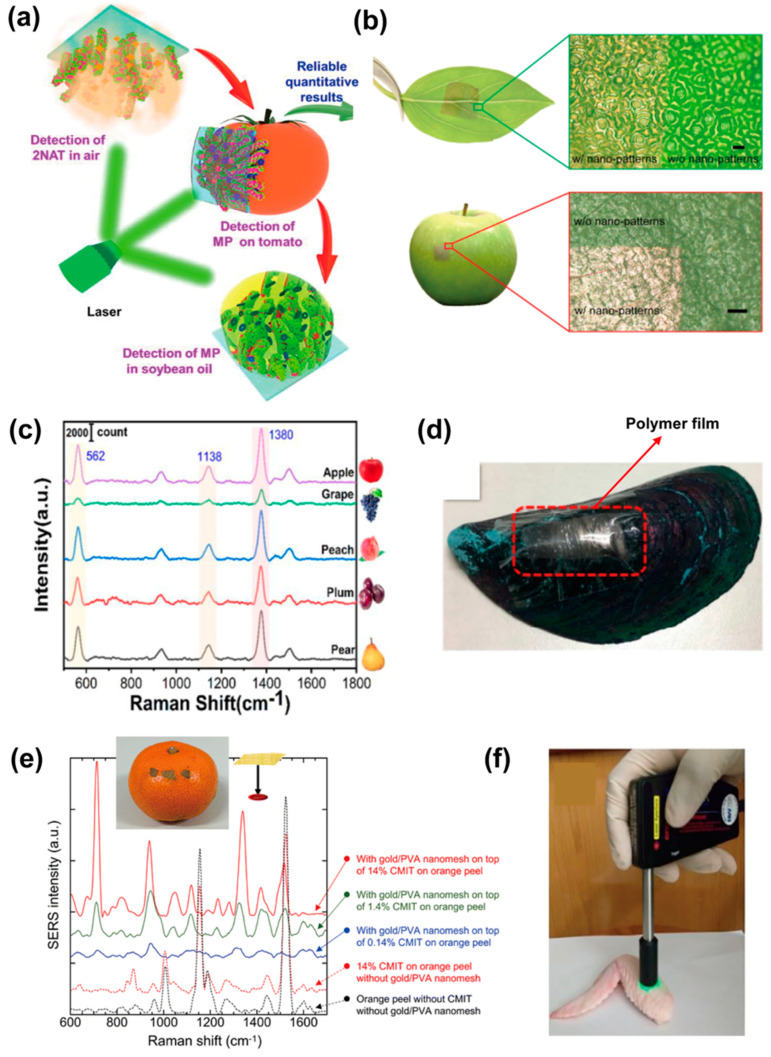
Detection of pesticides by using stretchable SERS substrates. (**a**) Schematic of the application of Ag/Au NWs/PDMS films for pesticide testing [102]; (**b**) photography of basil leaves and apples covered with ultra-flexible gold nano-patterned film and respective surface morphology micrographs [60]; (**c**) SERS spectra of thiram on various fruits peels [110]; (**d**) schematic of contacting polymer film onto the green mussel [57]; (**e**) SERS spectra (solid curves) of an aqueous solution of CMIT at various concentrations on the bottom surface of the gold/PVA nanomesh substrate on an orange peel [47]; (**f**) in situ detection of CPX and CAP on the chicken wing surface using the flexible Au−NSs/PMMA SERS substrate by the 532 nm excitation [112].

**Figure 10 nanomaterials-13-02968-f010:**
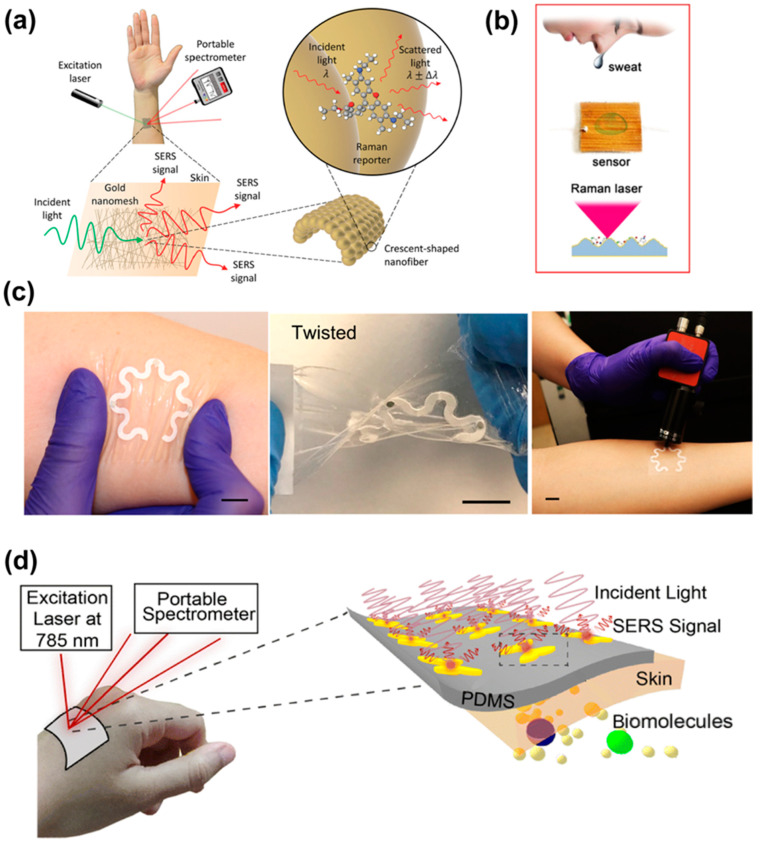
Stretchable SERS substrates for biomedical applications. (**a**) Concept of wearable SERS on the skin [118]; (**b**) a schematic of the multifunctional wearable electronic skin for the detecting human sweat [49]; (**c**) a schematic diagram of paper-based device twisting, deformation, and portable Raman measurement [120]; (**d**) a schematic diagram of a plasma exciton SERS sensor for detecting wrist sweat [56].

**Figure 11 nanomaterials-13-02968-f011:**
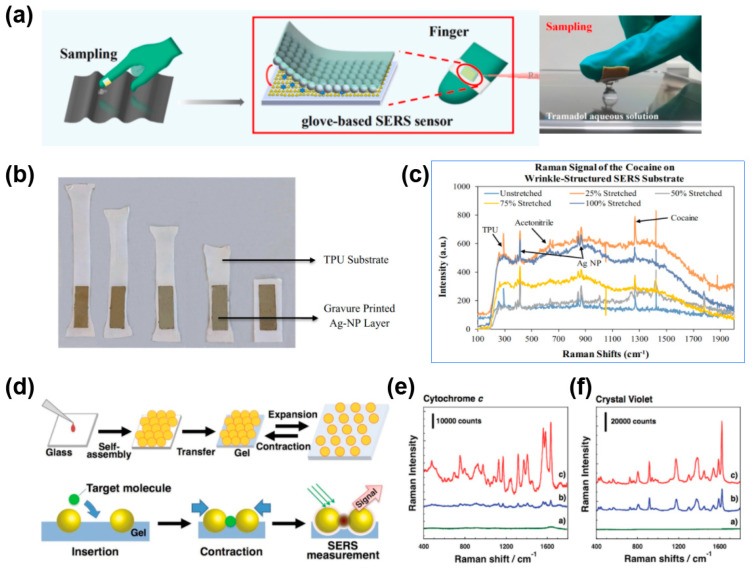
Other applications of stretchable SERS substrates. (**a**) Schematic diagram of the glove-based SERS sensor [124]; (**b**) a schematic illustration of the fabrication of SERS substrate using gravure printing process and (**c**) Raman signal of cocaine detected by the wrinkle−structured SERS substrate [125]; (**d**) preparation of a tunable plasmonic nanostructure and its applications for (**e**) *cytochrome c* and (**f**) crystal violet detection, where a) no gap control as “open form”—the approach where analytes were injected onto a gel in an expanded state (in MilliQ water) and analyzed as it is; b) no gap control as “closed form”—the approach where analytes were injected onto a gel in a contracted state (in 1 M NaCl solution) and analyzed as it is; and c) active gap control as “open−to−closed form”—the approach that target molecules were injected onto a gel is an expanded state (in MilliQ water) and analyzed after contraction (in 1 M NaCl solution) [126].

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
