# Peer review of "Recent Development and Applications of Stretchable SERS Substrates"

_nanomaterials, 2023, doi:10.3390/nano13222968_

Round 1
Reviewer 1 Report
Comments and Suggestions for Authors
Comments on the Quality of English LanguageAlready in the uploaded file
Author Response
We thank the reviewer for the valuable comments. Please see the attachment for the response letter.

Reviewer 2 Report
Comments and Suggestions for Authors
Authors summarized and discussed recent developments and applications of the stretchable surface-enhanced Raman scattering (SERS) substrates. in this review, authors summarized fabrication techniques of stretchable SERS substrates and described applications of stretchable SERS substrates in biosensing, environmental monitoring, and chemical analysis.
However, It is required to describe compatibility between sensing medium and stretchable SERS substrates
Comments on the Quality of English Language
Author Response

(The authors gave the same response as above.)

Reviewer 3 Report
Comments and Suggestions for Authors
The paper was well written and can contribute to SERS research community. I have a few minor comments.
The authors did not show the copy right permission to reuse figures. Please confirm that the authors got permission to reuse figures from each journal.
The brief theoretical description of SERS mechanism would help readers using important some equations.
Author Response

(The authors gave the same response as above.)

Round 2
Reviewer 1 Report
Comments and Suggestions for Authors
I have appreciated the extensive editing made by the authors. Now the paper is readeable and of good quality.